**Carbonic anhydrase is involved in calcification by the benthic foraminifer *Amphistegina***
***lessonii***
Siham de Goeyse[1], Alice E. Webb[1], Gert-Jan Reichart[1,2], Lennart J. de Nooijer[1]
[1] *Department of Ocean Systems, NIOZ Royal Netherlands Institute for Sea Research and Utrecht University, Texel,*
*Netherlands*
[2] *Department of Earth Sciences, Faculty of Geosciences, Utrecht University, Utrecht, Netherlands*
*corresponding author: siham.de.goeyse@nioz.nl
Key words: Foraminifera, calcification, symbiont, photosynthesis, carbonic anhydrase
**Abstract**
Marine calcification is an important component of the global carbon cycle. The mechanism by which some
organisms take up inorganic carbon for the production of their shells or skeletons, however, remains only partly
known. Although foraminifera are responsible for a large part of the global calcium carbonate production, the
process by which they concentrate inorganic carbon is debated. Some evidence suggests that seawater is taken up
by vacuolization and participates relatively unaltered in the process of calcification, whereas other results suggest
the involvement of transmembrane transport and the activity of enzymes like carbonic anhydrase. Here, we tested
whether inorganic carbon uptake relies on the activity of carbonic anhydrase using incubation experiments with
the perforate, large benthic, symbiont-bearing foraminifer *Amphistegina lessonii*. Calcification rates, determined
by the alkalinity anomaly method, showed that inhibition of carbonic anhydrase by acetazolamide (AZ) stopped
most of the calcification process. Inhibition of photosynthesis by either 3-(3,4-Dichlorophenyl)-1,1-dimethylurea
(DCMU) or by incubating the foraminifera in the dark, also decreased calcification rates, but to a lesser degree
than with AZ. Results from this study show that carbonic anhydrase plays a key role in biomineralization of
*Amphistegina lessonii* and indicates that calcification of those perforate, large benthic foraminifera might, to a
certain extent, benefit from the extra DIC which causes ocean acidification.
**1 Introduction**
Fossil fuel burning and land use changes have been steadily increasing atmospheric $CO_2$ levels. About $1/3^{rd}$ of the
added carbon has been taken up by the ocean (Sabine and Tanhua, 2010) and the resulting increase in seawater
dissolved carbon dioxide and associated acidification are lowering the saturation state of sea water with respect to
calcite and hence likely affects marine calcifiers. Even a modest impact on the production of carbonate shells and
skeletons may have important consequences for the global carbon cycle. Foraminifera are responsible for almost
25% of the total marine calcium carbonate production (Langer, 2008) and their response to ongoing acidification
is therefore important to predict future marine inorganic carbon cycling. Despite its relevance for future $CO_2$
scenarios, it is still unclear how increased $p\mathrm{CO}_2$ in seawater will affect foraminiferal calcification. Previous
research has shown discrepancies in their results: in some cases a higher $pCO_2$ increased the growth rate of benthic
foraminifera, while in other cases calcification decreased or halted (Haynert et al., 2014; Hikami et al., 2011).
Addition of $CO_2$ to sea water not only reduces saturation state with respect to calcite but also increases the total
dissolved inorganic carbon (DIC) concentration. At surface seawater pH, the dominant DIC species is $HCO_3^-$ and
many marine calcifiers are shown to employ transmembrane bicarbonate ion transporters (e.g. coccolithophores
(Brownlee et al., 2015; MacKinder et al., 2011); scleractinian corals (Cai et al., 2016; Giri et al., 2019; Zoccola et
al., 2015)), which may also be the case for foraminifera. If so, ocean acidification would be detrimental as this
shifts the carbonate system from $HCO_3^-$ to $CO_2$. Alternatively, $CO_2$ may be the inorganic carbon source of choice
for benthic foraminifera, as it diffuses relatively easily through lipid membranes. The latter uptake mechanism
would facilitate foraminiferal calcification as ongoing $CO_2$ dissolution increases total DIC and hence the
availability of building blocks for chamber formation. Since this uptake mechanism is crucial for calcification in
a rapidly changing ocean and because it is essentially unknown how foraminifera take up inorganic carbon, it
remains difficult to predict the reaction of foraminifera to ongoing environmental change. It was recently suggested
that $CO_2$ uptake by benthic foraminifera is achieved through proton pumping (Glas et al., 2012; Toyofuku et al.,
2017). The outward proton flux increases the $pCO_2$ directly outside the site of calcification (SOC) through
conversion of bicarbonate into carbon dioxide. The elevated pH at the foraminifers' site of calcification (Bentov
et al., 2009; de Nooijer et al., 2009) and reduced pH outside the cell thus results in a strong inward-outward $pCO_2$
gradient, promoting inward $CO_2$ diffusion. If calcification in foraminifera relies on this inward $CO_2$ diffusion, the
conversion from $HCO_3^-$ outside the test may be a limiting step for ongoing calcite precipitation. This process may
be catalyzed by an enzymatic conversion by carbonic anhydrase (CA), which is present in many prokaryotes and
virtually all eukaryotes (Hewett-Emmett and Tashian, 1996; Lionetto et al., 2016). This enzyme is essential in
calcification in many organisms, including corals, sponges and coccolithophores (Bertucci et al., 2013; Medaković,
2000; Müller et al., 2013; Le Roy et al., 2014; Wang et al., 2017). Also for foraminiferal calcification it has been
hypothesized that CA is used to enhance inorganic carbon uptake. Indirect evidence for such a role in calcification
comes from the observed slope between the carbon and oxygen isotopes (Chen et al., 2018), but direct evidence
is, however, still missing.

To test whether carbonic anhydrase is involved in biomineralization of perforate, benthic foraminifera we
incubated calcifying specimens of *Amphistegina lessonii* with acetazolamide (AZ), a membrane-impermeable
inhibitor of this enzyme (Elzenga et al., 2000; Moroney et al., 1985). Calcification and respiration were determined
by measuring changes in alkalinity and DIC of the incubated seawater over the course of the experiment. An
additional experiment was conducted in parallel to test whether CA is directly involved in perforate foraminiferal
calcification or that the effect is indirect via photosynthesis. The latter would imply that CA drives photosynthesis
by the symbionts and that observed effects would be due to reduced photosynthesis impairing calcification through
reduced energy transfer from the symbionts to the foraminifer.
**2 Material and methods**
**2.1 Foraminifera and incubations**
Surface sediments were collected from the Indo-Pacific Coral reef aquarium in Burgers' Zoo (Arnhem, the
Netherlands; Ernst et al., 2011). The sediments were kept at 24 ºC, with a day/night cycle of 12h/12h. Living

specimens of *Amphistegina lessonii* showing a dark cytoplasm and pseudopodial activity were manually selected, using a fine brush under a stereomicroscope and transferred to Petri dishes. They were fed with freeze-dried *Dunaliella salina* and incubated in North Atlantic sea water (salinity: 36). After a week, viable specimens were collected and divided over eight experimental conditions, each of them consisting of three groups (Fig. 1). Each group consisted of 40-60 specimens with a similar size distribution (initial diameter: 140 to 1200 μm, shown in S1). Foraminifera were placed in air-tight glass vials of 80 ml (24°C, 12h day-light cycle) for 5 days. Illumination was approximately 180 μmol photons $m^{-2}$ $s^{-1}$, during the 12h of light.

In the first experiment, the impact of acetazolamide (AZ) on calcification was tested. A stock solution was prepared by dissolving AZ (Sigma-Aldrich) in dimethyl sulfoxide (DMSO; 0.05% v/v) at a final concentration of 90 mM. It has been shown that DMSO at concentrations of 10-20% v/v does not impair calcification (Moya et al., 2008), so that the effect of this solvent is not reported here separately. The AZ stock solution was diluted with seawater from North Atlantic to achieve AZ concentrations of 4, 8 and 16 μM, which were used to incubate the foraminifera in. In a second experiment, inhibition of photosynthesis was tested by 1) addition of 3-(3,4-Dichlorophenyl)-1,1-dimethylure (DCMU ; Tóth et al., 2005; Velthuys, 1981) and 2) darkness. DCMU was added to seawater at a final concentration of 6 μM, whereas covering the vials with aluminum foil prevented light-dependent reaction and hence photosynthesis in a second set of incubations (Fig. 1).

**2.2 Alkalinity, DIC and nutrient analysis**

To quantify calcification and respiration, total alkalinity ($T_A$) and the concentration of dissolved inorganic carbon [DIC] were determined at the beginning and end of every incubation. This method was chosen above other growth method measurements such as sample weighing or counting chamber addition as it allows a quantification of the amount of calcite formed during the actual experiment. Total alkalinity was analyzed immediately at the end of each experiment, whereas subsamples to determine nutrient concentrations and DIC analyses were stored at -20°C (nutrients) and 4°C (DIC). The samples for DIC analyses were poisoned with mercury chloride (DIC) until analysis. These samples first passed a 0.2 μm syringe filter.

Alkalinity measurements were performed using an Automated Spectrophotometric Alkalinity System (ASAS), as described by Liu et al. (2015). Briefly, 60 mL of seawater are placed in a borosilicate vial and automatically titrated with a solution of 0.1 M HCl. Before the start of the titration, 45 microliters of bromocresol purple (10 mmol/L) was added to the seawater and pH changes were followed by spectrophotometry. Certified reference material (CRM; Dr. Dickson, Scripps Institution of Oceanography) was analyzed at the beginning of every series (5-10 samples) of measurements. Reproducibility of the obtained $T_A$ was ~3 μmol/kg (SD), based on 50 measurements of untreated seawater.

Nutrient samples were analysed on a QuAAtro continuous flow analyzer (SEAL Analytical, GmbH, Norderstedt, Germany) following GO-SHIP protocol (Hydes et al., 2010). DIC was measured on an autoanalyzer TRAACS 800 spectrophotometric system as described in Stoll et al. (2001).

**2.3 Calcification rate**

Changes in DIC and alkalinity between start and end of the experiments were used to calculate the net respiration
and calcification (Fig. 2). Total measured alkalinity is defined as the contribution of the following anions:
$T_{Ameasured} = [HCO_3^-] + 2[CO_3^{2-}] + [OH^-] + 3[PO_4^{3-}] + [HPO_4^{2-}] + [NO_3^-] - [H^+] - [NO_4^+]$          (1)
Concentrations of boron and silicon were neglected as the first one is constant and the second one is present at a
low abundance. In order to account for the alkalinity change related to the inorganic carbon system only, we
subtracted the combined concentrations of the nutrients from the measured alkalinity so that the observed alkalinity
over time is defined as:
$T_A = [HCO_3^-] + 2[CO_3^{2-}] + [OH^-] - [H^+]$          (2)

$Resp_{net}$ is defined as the difference between respiration and photosynthesis. Here, we consider the respiration of
the holobiont (foraminifera and its symbionts), which is calculated by:
$Resp_{net} = \Delta DIC - \Delta T_A / 2$          (3)


Since other processes, e.g. respiration by bacteria, may affect the $T_A$ and [DIC] during the incubations, vials were
carefully checked for the presence of biofilms. There was no sign of such activity in any of the treatments, so
changes in $T_A$ and [DIC] are attributed to the foraminifera and their symbionts.
**3 Results**
**3.1 Carbonic anhydrase inhibition**

Without acetazolamide, $T_A$ decreased on average by 53 µmol.kg$^{-1}$ and DIC by 38 µmol.L$^{-1}$ during the incubation
(table 1). This corresponds to 2.74 g/L of precipitated calcite. Contrastingly, when the seawater contained
acetazolamide (even at the lowest concentration of 4µM), alkalinity and DIC did not change or decreased only
marginally during the incubation (less than 0.4 g/L of calcite precipitated). When comparing the changes in $T_A$
and DIC between treatments, calcification is minimized by the AZ and net respiration slightly increases (Fig. 3).
The concentration of AZ has no discernible effect on the magnitude of changes in calcification/ respiration.

The number of chambers added by the foraminifera shows that the average number of chambers added decreases
after addition of AZ (table 3). Whereas many specimens in the control vials added 2 chambers, almost all
calcification after addition of AZ resulted in the addition of only one chamber.

**3.2 Photosynthesis inhibition**

When photosynthesis was not impaired (light control), alkalinity decreased within the vials by 70 µmol·L$^{-1}$ and
DIC increased by 21 µmol·L$^{-1}$ (table 2). Given the relative standard deviations, this is similar to the changes in $T_A$
and DIC in the control vials for the AZ-experiments. These changes correspond to approximatively 3.75 g·L$^{-1}$ of
precipitated calcite. In contrast, when foraminifera were cultivated in the dark or in presence of the photosynthesis
inhibitor DCMU, DIC increased by 16 and 42 $\mu$mol$\cdot$L$^{-1}$, respectively whereas the total alkalinity decrease was
only 19 and 11 $\mu$mol$\cdot$L$^{-1}$, respectively, which corresponds to less than a third of the amount of calcite precipitated
when photosynthesis was not hampered (Fig. 4). Changes in DIC/ $T_A$ are also reflected in the number of chambers
added to the incubated foraminifera: with DCMU or AZ added and in the dark, specimens added less chambers
than the control group (table 3). Some of the smaller specimens incubated during the experiment were not retrieved
from the vial, explaining the missing specimens (table 3). The foraminifera incubated with an inhibitor have more
broken chambers than the others.

**4. Discussion**
**4.1 Growth rates and the effect of AZ**
In the control experiments (incubations with unaltered seawater), foraminiferal calcification resulted in a decrease
in alkalinity of the culture media by approximately 53 $\mu$mol$\cdot$L$^{-1}$ over a period of 5 days (table 1). On average, this
equals a growth rate of 1.0 $\mu$g$\cdot$Ind.$^{-1}\cdot$day$^{-1}$, which is low when compared to some previously reported rates for
foraminiferal calcification (~6-60 $\mu$g$\cdot$Ind.$^{-1}\cdot$day$^{-1}$; (Evans et al., 2018; Glas et al., 2012; Keul et al., 2013). These
studies, however, all used different species than the one incubated here. Previous research using *Amphistegina* spp.
reported growth rates of 3-9 and 2.6-4 $\mu$g$\cdot$Ind.$^{-1}\cdot$day$^{-1}$ (ter Kuile and Erez, 1984; Ter Kuile and Erez, 1987),
respectively, while Hallock et al. (1986) reported rates of 0.3-6.6 $\mu$g$\cdot$Ind.$^{-1}\cdot$day$^{-1}$ depending on the light intensity.
Segev and Erez (2006) reported growth rates similar to those observed in our study (0.53-1.0 $\mu$g$\cdot$Ind.$^{-1}\cdot$day$^{-1}$),
based on changes in dry weight. The growth rates reported here fall in the lower range of those previously reported,
which may be due to the average size of our specimens, the used light intensity and/ or the short duration of our
experiment.

Addition of AZ caused a 20 fold decrease in calcification rates (Fig. 3), while increasing net respiration. The
concentration of the inhibitor (4-16 $\mu$M) did not affect the magnitude by which net calcification decreased, nor
does it appear to affect the increase in net respiration (Fig. 3). The accompanying decrease in the number of
chambers added per specimen (table 3), suggests that AZ did not decrease the survival rates of the incubated
specimens, but affected the rate of chamber addition in all specimens equally. The inhibition of calcification caused
by AZ suggests that carbonic anhydrase plays a crucial role in perforate foraminiferal biomineralization. With the
inhibitor present, specimens produced little to no calcite (Fig. 3), indicating that either biomineralization relies on
CA, or is negatively impacted through an effect of CA on photosynthesis. Whether calcification depends directly
on extracellular carbonic anhydrase (eCA) or that calcification depends on photosynthesis and thereby indirectly
on CA, can be inferred from comparing the two sets of experiments (Fig. 1).

**4.2 Effect of photosynthesis on calcification**
The inhibition of photosynthesis with DCMU and darkness decreases calcification comparably (Fig. 3).
Simultaneously, net respiration increases after addition of DCMU, and so does blocking light (Fig. 4). The
similarity in the effect of darkness and DCMU indicates that photosynthesis has an effect on calcification in these
perforate foraminifera. It was previously suggested that light, irrespective of photosynthesis, enhances calcification
in foraminifera (Erez, 2003). Since the latter study used the planktonic, low-Mg calcite *Globigerinoides sacculifer*,
the discrepancy between results may be caused by differences in the process involved in calcification between
these species. For example, it has been suggested that calcification may involve seawater transport (Erez, 2003;
Segev and Erez, 2006) as well as transmembrane transport (Nehrke et al., 2013; Toyofuku et al., 2017), of which
the relative contribution may vary between groups of foraminifera.
Foraminiferal calcification and endosymbiont photosynthesis both require inorganic carbon. Therefore, it seems
reasonable to suggest that those two mechanisms are competing with each other for inorganic carbon, as was
shown by Ter Kuile et al. (1989b, 1989a). However, our results show that preventing photosynthesis by the
symbionts actually decreases foraminiferal calcification. This implies that benefits from photosynthesis overcomes
an eventual competition with calcification, which is in agreement with results from Duguay (1983) and Hallock
(1981) who showed that both calcium- and inorganic carbon uptake into the cell is enhanced by light. As the
foraminifera were in the dark 12h hours a day it might also be that DIC is shared over time, being used for
calcification during the dark phase and $CO_2$ being used for calcification during the light phase.
It was shown that photosynthetic symbionts provide energy to their foraminiferal hosts (Lee, 2001) and that
calcification in some foraminifera is enhanced by the photosymbiont's activity (e.g. Hallock, 2000; Stuhr et al.,
2018). This was for example seen already by Muller (1978), reporting increased carbon fixation by the foraminifer
*A. lessonii* in the light compared to uptake of carbon in the dark. A positive effect of higher $CO_2$ level on
calcification through enhanced photosynthesis is known as "fertilization effect" (Ries et al., 2009). A positive
effect of photosynthesis on calcification has been observed previously for other marine calcifyers as well. For
example, in coccolithophores, decreasing $CO_2$ can hamper calcification through reduced photosynthesis
(Mackinder et al., 2010). Utilization of photosynthate as an organic template for calcification may explain this
observation. We here hypothesize that a similar effect may explain decreased calcification in foraminifera as a
consequence of inhibited photosynthesis (Fig. 3), as hypothesized by Toler and Hallock (1998). If so, the type of
organic molecules produced by the foraminifer's endosymbionts and their fluxes will need to be assessed to test
the extent of the dependency of calcification on photosynthesis. However, it has been shown that symbiotic
dinoflagellates can trigger the activity of carbonic anhydrase from their host organisms (giants clams and sea
anemones) (Leggat et al., 2003; Weis, 1991; Weis and Reynolds, 2002; Yellowlees et al., 2008), thereby explaining
how photosynthesis enhances calcification. Alternatively, increased activity of CA in the symbiont may also
promote the flux of products to the host and thereby promote calcification indirectly. Since there are many
(perforate) foraminiferal species that do not have photosynthetic symbionts, the effect of inhibiting CA in these
species may provide additional information on the role played by CA in calcification.

**4.3 Role of CA in calcification**
In calcifyers other than foraminifera, carbonic anhydrase plays a direct role in calcification. For example, in giant
clams (Chew et al., 2019), gastropods (Le Roy et al., 2012) and oysters (Wang et al., 2017), CA helps to concentrate
inorganic carbon in the fluid from which calcium carbonate precipitates. In scleractinian corals, CA promotes
conversion of metabolic $CO_2$ into bicarbonate after the carbon dioxide diffused into the sub-calicoblastic space
(Bertucci et al., 2013). Although the inorganic carbon would take the same route in absence of CA, the hydration
of $CO_2$ is relatively slow and ion fluxes and calcification rates would be a fraction what they are with the catalytic
activity of CA. This role of CA fits with the localization of (membrane-bound) CA observed at the walls of the
calicoblastic cells by immunolabelling (Moya et al., 2008). In addition, by facilitating an inward flux of inorganic
carbon, involvement of CA can explain the co-variation of oxygen and carbon isotopes in coral aragonite (Chen
et al., 2018; Uchikawa and Zeebe, 2012).
In larger benthic foraminifera, CA likely plays different roles: it helps concentrating $CO_2$ by the symbionts and
aids foraminiferal calcification. It is also likely that cytoplasmic CAs -involved for instance in intracellular pH
regulation- also affect calcification. The molecular types of CA that are involved and their precise location still
remain to be investigated within the larger benthic foraminifera. In addition, the type of symbionts or their absence,
may affect inorganic carbon uptake, so that the result obtained here may only partially apply to foraminifera in
general. Analogous to other calcifying organisms and based on existing models of foraminiferal calcification, we
hypothesize that extracellular CA helps to convert $HCO_3^-$ into $CO_2$ directly outside the calcifying chamber. This
would help to further increase the $p$$CO_2$ outside the foraminifer in addition to the shift in inorganic carbon
chemistry resulting from active proton pumping and subsequent low pH (Glas et al., 2012; de Nooijer et al., 2009;
Toyofuku et al., 2017). Although not directly targeted by our experimental approach, as the inhibitor we used is
membrane impermeable, it is likely that a form of CA within the calcifying fluid increases the rate by which the
diffused $CO_2$ is converted into bicarbonate.
The involvement of extracellular CA in calcification may explain why perforate foraminifera can be relatively
resilient to ocean acidification. It also remains to be investigated whether Tubothalamea, who produce their calcite
in a fundamentally different way (Mikhalevich, 2013; Pawlowski et al., 2013), use CA similarly. If they rely on
CA for conversion of $HCO_3^-$ to $CO_2$ and take up inorganic carbon by diffusion of $CO_2$, additional dissolved
atmospheric $CO_2$ may be beneficial for calcification in foraminifera. If they exclusively rely on bicarbonate ions,
a reduction in pH would lower the $[HCO_3^-]$ and thereby hamper calcification. Manipulation of the inorganic carbon
speciation in relation to calcification and the aid of enzymes therein, will allow predicting rates of calcification as
a function of ongoing ocean acidification.

## 5 Conclusions

The alkalinity anomaly method allowed us to quantify growth rates in incubation experiments, equalling addition
of 1 µg/individual/day. Calcification and photosynthesis in the benthic foraminifer *Amphistegina lessonii* and its
symbionts both depend on carbonic anhydrase (CA) as shown after inhibition by acetazolamide (AZ). Since the
inhibitor is membrane-impermeable, the CA may well be localized at the outside of the foraminifer's cell
membrane. Our results also show that inhibiting photosynthesis by DCMU or incubation in darkness reduce
calcification similarly. This suggests that not light, but photosynthesis itself promotes calcification in symbiont-
bearing perforate foraminifera. We also suggest that CA plays a role in concentrating inorganic carbon for
calcification, possibly by promoting conversion of bicarbonate into carbon dioxide outside the foraminifer.

## Data availability

The data on which this publication is based can be found through the following DOI: 10.4121/uuid:afcdcdc1-2591-
4822-bade-806119cdd724

Authors contribution:
SdG and LJdN designed the experiment and SdG carried it out. SdG and AEW analysed the seawater inorganic
chemistry. SdG and LJdN analysed the data and prepared the manuscript with contributions from all co-authors.

**Competing interests**

The authors declare they have no conflict of interest

**Acknowledgments**

We would like to thank Karel Bakker for DIC measurements. We kindly thank Max Janse (Burgers' Zoo, Arnhem)
for providing stock specimens of *Amphistegina lessonii* and Kirsten Kooijmans and Michele Grego (NIOZ) for
providing cultures of Dunaliella salina. This study was also carried out under the program of the Netherlands Earth
System Science Centre (NESSC; 024.002.001), financially supported by the Dutch Ministry of Education, Culture
and Science (OCW). LdN acknowledges financial support from the NWO open competition program (Pasodoble
Grant). We thank Takashi Toyofuku, the three anonymous reviewers and the editors for their helpful comments
and all subsequent improvements.

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

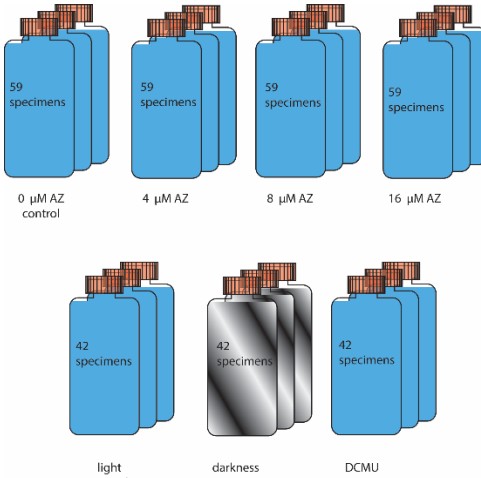

Figure 1: 59 specimens were placed in one culture vial, with three replicate vials for each concentration of acetazolamide (upper row). Similarly, 42 specimens were incubated under light, in the dark and with the inhibitor DCMU (lower row).

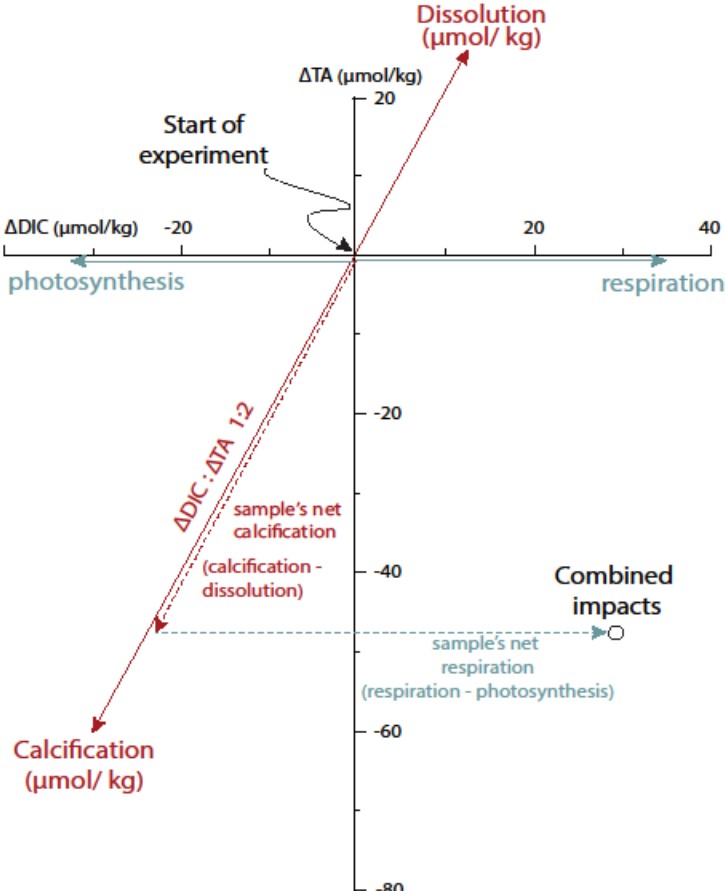


**Figure 2: Calcification and net respiration of foraminifera deduced from changes in DIC and total alkalinity over time.**
**The blue vectors show the impact of photosynthesis and respiration (impacting DIC), the red arrow show the impact of**
**calcification and calcite dissolution (impacting both DIC and TA in a 1:2 ratio). Observed changes for each incubation**
**should be decomposed into two vectors: a contribution of calcification (dashed red arrow) and the net effect of**
**respiration and photosynthesis (dashed blue arrow). Approach is indicated here for a hypothetical incubation).**

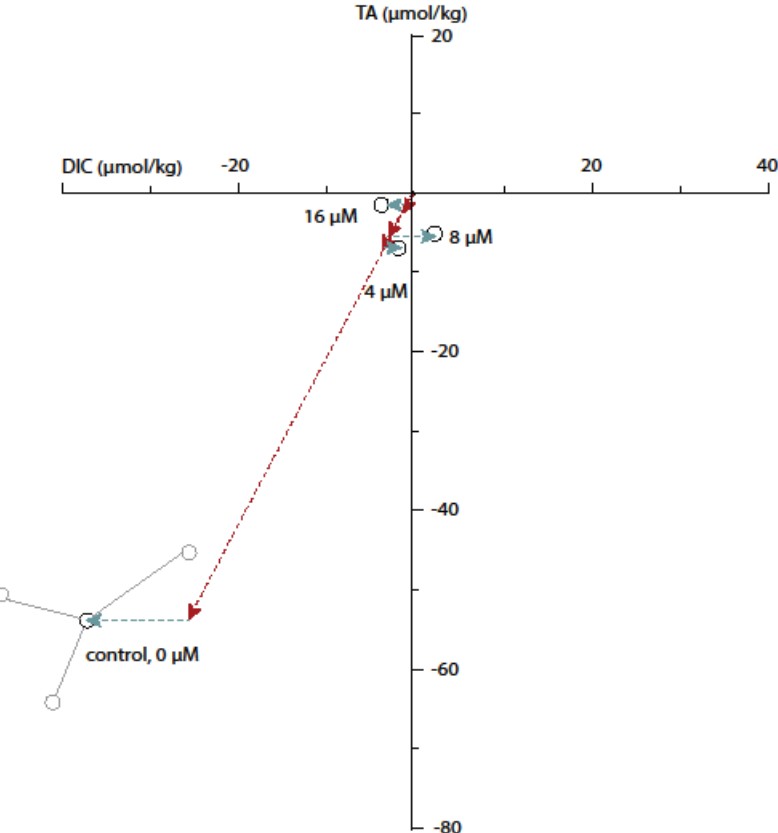


**Figure 3: Changes in total alkalinity versus DIC for all concentrations of acetazolamide (AZ) used. Every black circle represents the average change in DIC-T$_A$ for one triplicate of incubations. The three grey circles show the measured DIC/ T$_A$ combination for each of the triplicate measurements within the control treatment. For the three additions of AZ, replicates never differed more than 8 µmol/kg from the average for DIC and never more than 5 µmol/kg from the average for T$_A$. Arrows show the calcification (red) and net respiration (blue) effects.**

453

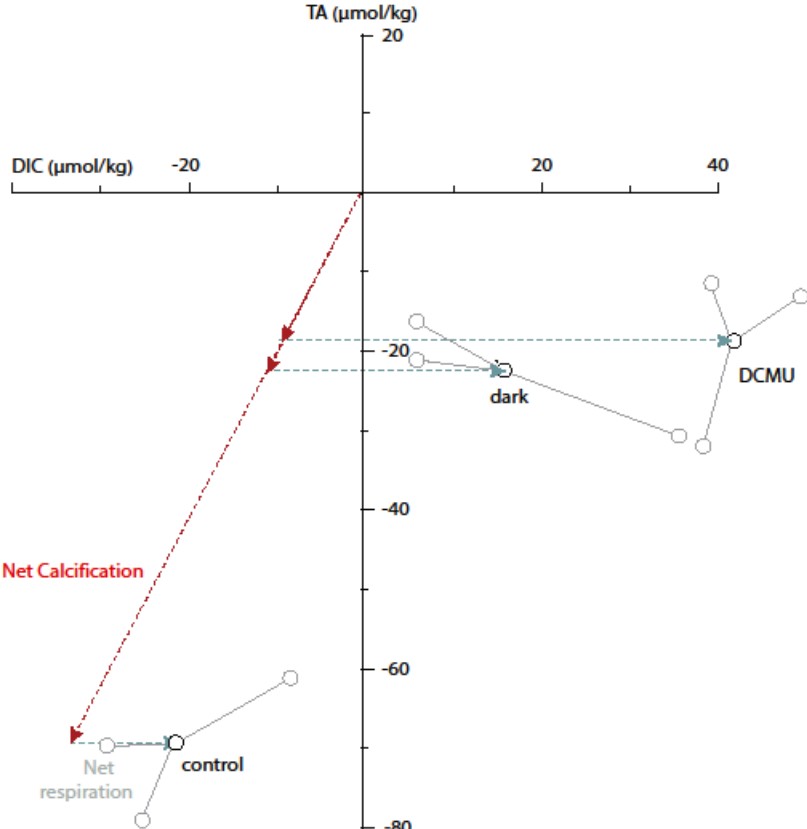

454

**Figure 4: Changes in total alkalinity versus that in DIC for incubations in light-dark alternation (control), in the dark and with the photosynthetic inhibitor DCMU. Every black circle represents the average change in $T_A$ and DIC between the initial and the final values for each triplicate. The three grey circles show the measured DIC/ $T_A$ combination for each of the triplicate measurements within every of the three treatments. For the 'dark' and 'DCMU' treatments, the individual DIC/$T_A$ combinations are connected to the average value. Arrows show the calcification (red) and net respiration (blue) effects.**

461

| [AZ] (μM) | Initial $T_A$ | Δ $T_A$ | Initial DIC | Δ DIC |
|---|---|---|---|---|
| 0 | 2284 | - 53± 8 | 2110 | -38 ± 9 |
| 4 | 2285 | -7 ± 1 | 2105 | -2 ±2 |
| 8 | 2285 | -5 ± 1 | 2105 | 3 ± 7 |
| 16 | 2292 | -2 ± 4 | 2109 | -3 ±6 |

**Table 1: Total alkalinity and DIC changes for every triplicate. Confidence interval: 1 STD (taking biological variability into account)**

| Vial | Initial $T_A$ | Δ $T_A$ | Initial DIC | Δ DIC |
|---|---|---|---|---|
| control | 2280 | -70 ±7 | 2115 | -21 ± 9 |
| DCMU | 2286 | -22 ±9 | 2091 | 42 ±14 |
| dark | 2280 | -19 ±6 | 2115 | 16 ±5 |

**Table 1 : Total alkalinity and DIC changes for every triplicate. Confidence interval: 1 STD (taking biological variability into account)**

| Experiment | Total no of specimens incubated | Number of specimens that added: | | | |
|---|---|---|---|---|---|
| | | 1 chamber | 2 chambers | 3 chambers | 4 chambers |
| AZ, 0 μM | 80 | 25 | 19 | 1 | 1 |
| AZ, 4 μM | 100 | 17 | 4 | 0 | 0 |
| AZ, 8 μM | 123 | 15 | 2 | 0 | 0 |
| AZ, 16 μM | 135 | 6 | 0 | 0 | 0 |
| control, light | 123 | 40 | 25 | 1 | 0 |
| DCMU | 115 | 16 | 1 | 0 | 0 |
| dark | 122 | 18 | 0 | 0 | 0 |

**Table 3: Number of chambers added per specimen for each treatment**