# Peer review of "Carbonic anhydrase is involved in calcification by the benthic foraminifer Amphistegina 1 2 lessonii"

_Biogeosciences, 2019_

## Referee Comment (RC1) · Anonymous Referee #1 · 14 Oct 2019

Assessment of "Carbonic anhydrase is involved in benthic foraminiferal calcification"

1. Overview of the manuscript and basic assessment:

This manuscript addresses an important question regarding calcification in foraminifera: "Does carbonic anhydrase play a role in inorganic carbon uptake?" The authors address this question using a pair of experiments using probably the second-most widely studied benthic foraminifer, *Amphistegina lessonii*. This is a warm-temperate to tropical species that is abundant throughout the Indo-Pacific and which grows abundantly in some large-scale reef aquaria, which was the source of the experimental specimens. The results of the experiment support previous experimental work showing that *Amphistegina* spp. can live and calcify at elevated $pCO_2$ levels (e.g., Glas et al., 2012; McIntyre-Wressnig et al., 2013; Knorr et al., 2017).

Unfortunately, the manuscript itself, while reporting interesting data, is not suitable for publication as currently written. There are numerous deficiencies in statements and assumptions regarding foraminifera, methods descriptions, and referencing, that must be addressed to bring this manuscript to publication quality.

2. Specific deficiencies.
   a. Title: Because the paper is written with the assumption that this experiment represents "benthic foraminiferal calcification", an erroneous assumption that will be addressed next, please change the title to" Carbonic anhydrase is involved in calcification in *Amphistegina lessonii*, a benthic foraminifer that hosts diatom endosymbionts". *Recommendation*: Change the Title.
   b. Introduction: A fundamental problem with the title and the paper overall is the inherent assumption that calcification in *A. lessonii* represents calcification in the benthic Foraminifera. While at least some of the co-authors know that is not a valid assumption (e.g., de Nooijer et al, 2009), the manuscript should at least make the distinction between calcification in Globothalmea (in this case, a hyaline, perforate foram) and Tubothalmea (imperforate, porcelaneous forams). This distinction is important because, as shown by Pawlowski et al. (2013) together with Mikhalevich (2014), these two groups evolved calcification independently. Moreover, since *Amphistegina* spp. host diatom endosymbionts, the carbonic anhydrase could be associated with the diatoms, in which case, the observations would not apply to hyaline taxa that do not host algal symbionts. *Recommendation*: Revise the Abstract, Introduction and Discussion to note that this experimental study applies to hyaline forams hosting diatom symbionts.
   c. Methods: There are many studies in the literature that discuss culture of *Amphistegina* spp. and *Heterostegina depressa*, as well as other benthic forams that host algal symbionts. The authors do not mention two important culture parameters, illumination (i.e., light intensities) and salinity. The latter may not be as critical to experimental results, since alkalinity is reported. However, light is a widely established, extremely important environmental parameter (e.g., Muller, 1978; Hallock, 1981; Hallock et al., 1986; Talge and Hallock, 2003; Williams and Hallock 2004). In addition, the authors do not report the size range of individuals

used in the experiments. If they had been aware of the earlier experimental studies, they would know that growth rates in *Amphistegina* are size dependent, which pertains to the comparability of their results to other studies (more on this in comments on the Discussion). Finally, in line 72, the authors mention that specimens were incubated in calcein prior to starting the experiment, with no mention of why and no further mention of calcien in the manuscript.
*Recommendations*: Please report the light environment of the cultures, the salinity of the culture media, and the approximate starting size (or ending, since the experiments were very short) of the experimental specimens. Also, either elucidate on the use of calcein or delete mention of it.

d. Results: The results are relatively straightforwardly presented. The only suggestion is that, in Table 1 and Table 2, reporting the decimal values for initial TA and initial DIC are not meaningful, given the standard deviations of the changes reflect whole numbers that represent ~10–40% of the changes in TA and DIC.

e. Discussion: See the comments and recommendations under "Introduction. That is, the Discussion should be focused on *Amphistegina* as a model for hyaline forams with algal endosymbionts, not all benthic forams.
Moreover, the authors state in lines 159–160, "The only previous study using *Amphistegina* spp."; that statement is inaccurate. Ter Kuile and Erez (1984, 1987); and Hallock et al. (1986), all reported rates of calcification in units equivalent to those reported. And indeed, the calcification rates reported in the submitted manuscript are lower than most of the previously reported rates for *Amphistegina*, which is why this reviewer questioned the light environment of the experiments.  If the light levels inside the culture flasks were limiting photosynthesis and growth of the experimental specimens, the calcification rates would of course be relatively low. See, for example, Table 1 in Hallock et al. (1986), who reported growth rates in μg/day dry weights at five different light intensities for both *A. lessonii* and *A. gibbosa*. The growth rates at the lowest light levels are similar to those reported in the submitted study. Moreover, the authors should note the starting diameters of the specimens used in the Hallock et al. (1986) paper. The experiment reported in Table 1 in that paper used recently produced juveniles, while the experiments reported in Table 3 included one trial with intermediate-sized specimens (500–600 um diameter), while the other trials also used small juveniles. If the experiments reported in the submitted paper used specimens in the 1–1.5 mm size range, the biology of the forams indicates that only a few specimens would have added new chambers.
Lines 177–179: The authors suggest that calcification in *Amphistegina* might differ from that reported in *G. sacculifer*, which is interesting, because, elsewhere, they are equating calcification in *Amphistegina* with calcification in miliolids, which are far more distantly related, as noted above.
In the paragraph in lines 187–196, the authors appear to assume that photosynthate produced by the algal symbionts is primarily used for organic matrix. They do not consider the production of simple sugars that can be used in ATP production that drives the ion pumps. In the case of *Amphistegina*, TEM studies have shown the abundance of lipid storage bodies in the vicinity of the

symbionts. Indeed, the authors' conclusion that more research is needed on the types of organic molecules produced is certainly true, but they overlooked pertinent information in papers by Lee, Stuhr, Talge, Toler, and probably others. They also overlooked pulse-chase studies by Muller (1978) and ter Kuile and Erez (1987).

In lines 193–194, the authors mention "symbiotic dinoflagellates and zooxanthellae". Zooxanthellae are symbiotic dinoflagellates.

*Recommendations*: See below.

f. Conclusions: Of course, it is photosynthesis that enhances growth and calcification in *Amphistegina*; and light is required for photosynthesis. The phylogenetic and physiological capability to calcify is inherent in the hyaline forams (that is why *Amphistegina* can exhibit some calcification in the dark). But because calcification is an energy-driven process, the substantial energy and organic matter provided by photosynthesis by algal symbionts substantially enhances growth, including calcification.

g. References: The references are typically "end-note" formatting-problematic and need extensive editing if *Biogeoscience* requires consistency in referencing. Examples:

Lines 259, 272, 288, 291, 303, 305, 316, 332, 335, 337, 228: genus and species names are not italicized

Lines 264, 290–291, 315–316, 335, 337–338: in manuscript titles, the nouns and some other words start with capital letters, inconsistent with referencing format for other journal articles.

Line 329 and 332, use subscripting, superscripting and Greek notation, as appropriate.

**Overall recommendations:** The authors should become much more familiar with the rather extensive literature on culture, growth, calcification, photosynthesis, physiological and cytoplasmic studies of *Amphistegina* spp. Then rewrite the entire paper, correcting misunderstandings, being more rigorous regarding what other taxa these observations may apply to, and incorporating appropriate citations. The experimental protocol and basic results appear to be sound and can be an important contribution to understanding calcification in hyaline forams that host algal endosymbionts. But the manuscript, as currently written, contains errors and misleading interpretations that detract substantially from the experimental results.

---

## Referee Comment (RC2) · Anonymous Referee #2 · 24 Oct 2019

General comments

The author present a set of experiments, performed on a high number of foraminifera specimen of a same common symbiotic species. In this experimental work, the effect of an extracellular inhibitor of the carbonic anhydrase (CA) enzyme is compared to the effect of a photosynthesis inhibitor, as well as the sole light deprivation. Biomineralisation change is evaluated through measurements of DIC concentration and alkalinity change, and using solely that approach to evaluate "biomineralisation yield" is also a main aspect of the article. These results evidence, in my opinion, the role of carbonic anhydrase, but I do believe that additional simple information should be given in order to confirm that no other phenomenon can explain, or interfere with, those results. If those information can be given (see below) and the role of CA is confirmed, then

[Figure]

the scientific significance of these results is excellent. The scientific quality is good, the method and experimental aspects are good despite the few information lacking, as great effort were provided to replicate the experiments and perform them on a significant amount of specimens. The discussion however and the manuscript text in general is not as good as I believe necessary for publication in an international journal. There are not enough references backing information, several aspects of the results are not discussed, a part of the discussion is just a description of results, there are words missing in some sentences, one name on a figure and a table do not match, one figure permitting the comparison of all results is missing, and there are several typography mistakes. I am not able to properly judge the english, but I found the manuscript perfectly understandable. If the text of the manuscript can be improved by the authors, I would recommend publication of the article as the results constitute a major advance in the understanding of biomineralisation by foraminifera (and in my opinion, it gives insight on biomineralisation mechanism in general considering how widespread is CA). For that reason, I hope the authors will improve the text, scientific content and discussion of the article in order to provide these interesting results the context they deserve to become a well referred to article.

Specific comments

**1 I understand that solely using chemical solution parameters to describe the evolution of biomineralisation is one of the suggestion of the article, I however believe it is not enough as some other parameters can affect DIC concentration and alkalinity: ex: microbial proliferation or open system phenomenon (improved gas /liquid phase exchanges in one experiment because of slightly different pH, or temperature differences due to the use of aluminium foil etc...). In my opinion the interest of a laboratory experiment on living organism cultured in vials is to be able to observe directly these organisms, which is not possible in other type of experiments. Are, in the end of the experiment, the vials clean enough with no particular microbial proliferation in one treatment? What proportion of foraminifera survived the experiment in all setups?**

How are the new chambers? The author used calcein, they should thus be able to image the new chambers formed in each media. I believe any experiment of that type should present some kind of imaging, or at least a description of the visual aspects of the experiment, validating that new chambers formed, and evaluating that no microbial proliferation could have explained death of several microorganisms, that could explain less biomineralisation. For example, in the acetazolamide experiment, if a microbial proliferation occurred and a third of the foraminifera died, while the other survived and biomineralised regularly, wouldn't it give the impression, just by measuring DIC and alkalinity variation that only biomineralisation was affected by acetazolamide? These are simple information that would strengthen the results and the methods, that should be provided in the manuscript before publication.

**2 In my opinion, authors should find a way to represent the results of all the different experiments together in one figure to ease comparison. As an example they could use the "corresponding g/L precipitated calcite" calculated for each experiment.**

**3 When discussing the effect of phostosynthesis on calcification (line 192) the author do not mention the effect of lowering ATP production and rather suggest that photosynthesis promotes the production of molecules that are used in organic templates of calcification. The role of ATP in chamber formation is, in my opinion, impossible to ignore, the author must discuss it in the manuscript. On the other hand, organics produced by the symbionts may help biomineralisation (this indeed need further investigation), but it should be mentioned that there are (many?) benthic foraminifera with a hyaline test that do not bear symbionts. This should be discussed by the authors as well.**

Technical corrections

Missing words or information:

L29: "saturation state" the author should specify that it is towards calcium carbonate

L35-36: The authors could specify the foraminifera species (benthic ? planktonic ? Amphistegina ?)

L44: "Since this uptake…." The sentence sounds odd, a word is probably missing, it should be rephrased.

L48: There is a dot after the bracket

L48: The sentence states "It was recently suggested that $CO_2$ uptake by foraminifera is achieved through proton pumping" is that correct or did the authors used a shortcut to say that proton pumping (and thus ATP consumption) is used to modify pH and thus favor $CO_2$ uptake/or that a proton pump is used to actively cotransport $CO_2$? This imprecision should be corrected.

L50: In my understanding Bentov paper rather says that $CO_2$ gets concentrated in low pH vesicles, and that from there it diffuses to the high pH vesicles where it converts into charged DIC species and is thus trapped inside the vesicle. The authors should clarify that point if I am not mistaking.

Line 112: I am not familiar with the "T" symbol signification next to each alkalinity species, could it be clarified?

L115: Is that equation calculating "the alkalinity" or "the change in alkalinity", this should be clarified. Figure 4 caption: there is a dot after "represents"

L193: the sentence is missing a word

L198 : "it has been shown that symbiotic dinoflagellates and zooxanthellae can trigger the activity of carbonic anhydrase (CA) in their host organisms […] thereby explaining how photosynthesis enhances calcification". The authors need to specify whether they mention the symbiont CA or if they refer to the host CA. Additionnally the link between CA and photosynthesis must be explained.

Missing references:

[Figure]

L27: "1/3rd of the carbon. . ." a reference should be provided

L40: Who suggested it? a reference is missing

L52: "many procaryotes and virtually all eukaryotes" a reference should be provided.

L60: " a membrane impermeable inhibitor of this enzyme" a reference must be added. (A reference attesting that DCMU inhibits photosynthesis should be added as well if not provided in the manuscript).

L207 : "Ca promotes [. . .] into the calcicoblastic space" this information should be supported by a reference

L214: this reference and thus probably the whole sentence (except if another reference can be given) must be suppressed (as mentioned on Biogeosciences website "Works cited in a manuscript should be accepted for publication or published already").

Other comments :

**1 Given that TA is measured with a $3\mu$mol/kg precision and given the errors given in the tables, the decimals should be suppressed. #2 L78: What is the final concentration of dimethyl sulfoxide in the final flasks? The effect of dimethyl sulfoxide at that concentration on foraminifera should have been checked in a control experiment, if not, it should at least be discussed. #3 Figure 3 and 4, error bars should be represented or mentioned in the caption if smaller than symbols, or, even better, each 3 point replicates could be represented. #4 Can the author explain why on figure 4 and 3 the control point is not at the same position (is there an explanation for these two different control results?). Additonnally, in table 1 and figure 3 there is two different names for one treatment, "No AZ" and "0 $\mu$M", please choose one wording. #5 From line 159 to 170 it is a summary of the results that should not be, in my opinion, in the discussion. #6 Line 175: The "extracellular" specificity of CA is mentioned here but not discussed in part 4.3 and then comes back later in the manuscript. This should be restructured to clarify the message of the auhtors. #7 Line 183: "the discrepancy between results**

may be caused by differences in the process involved in calcification between these species" Can the author mention one or more process they are referring to?

---

## Author Comment (AC1) · 25 Dec 2019

**Author responses to review of Anonymous Referee #1 to "Carbonic anhydrase is involved in benthic foraminiferal calcification"**
by Siham De Goeyse, Alice E. Webb, Gert-Jan Reichart, and Lennart J. de Nooijer

We are very grateful to the anonymous reviewer for the detailed comments on our manuscript. We reply below to the specific comments. Reviewer comments are given in italic font and our response in bold font.

OVERVIEW OF THE MANUSCRIPT AND BASIC ASSESMENT:

*"This manuscript addresses an important question regarding calcification in foraminifera: "Does carbonic anhydrase play a role in inorganic carbon uptake?" The authors address this question using a pair of experiments using probably the second-most widely studied benthic foraminifer, Amphistegina lessonii. This is a warm-temperate to tropical species that is abundant throughout the Indo-Pacific and which grows abundantly in some large-scale reef aquaria, which was the source of the experimental specimens. The results of the experiment support previous experimental work showing that Amphistegina spp. can live and calcify at elevated pCO2 levels (e.g., Glas et al., 2012; McIntyre-Wressnig et al., 2013; Knorr et al., 2017).*
*Unfortunately, the manuscript itself, while reporting interesting data, is not suitable for publication as currently written. There are numerous deficiencies in statements and assumptions regarding foraminifera, methods descriptions, and referencing, that must be addressed to bring this manuscript to publication quality."*

**We are very happy with the constructive comments by this reviewer and changed our manuscript accordingly. Below, we reply point-by-point and indicate what we have changed in the text of the revised version of our manuscript.**

SPECIFIC COMMENTS:

*"Title: Because the paper is written with the assumption that this experiment represents "benthic foraminiferal calcification", an erroneous assumption that will be addressed next, please change the title to" Carbonic anhydrase is involved in calcification in Amphistegina lessonii, a benthic foraminifer that hosts diatom endosymbionts". Recommendation: Change the Title. "*

**We agree with the reviewer that the original title may have been a bit too general. Therefore, we changed the title into: "Carbonic anhydrase is involved in calcification of the benthic foraminifer *Amphistegina lessonii*"**

*"Introduction: A fundamental problem with the title and the paper overall is the inherent assumption that calcification in A. lessonii represents calcification in the benthic Foraminifera. While at least some of the co-authors know that is not a valid assumption (e.g., de Nooijer et al, 2009), the manuscript should at least make the distinction between calcification in Globothalmea (in this case, a hyaline, perforate foram) and Tubothalmea (imperforate, porcelaneous forams). This distinction is important because, as shown by Pawlowski et al. (2013) together with Mikhalevich (2014), these two groups evolved calcification independently. Moreover, since Amphistegina spp. host diatom endosymbionts, the carbonic anhydrase could be associated with the diatoms, in which case, the observations would not apply to hyaline taxa that do not host algal symbionts. Recommendation:*

*Revise the Abstract, Introduction and Discussion to note that this experimental study applies to hyaline forams hosting diatom symbionts."*

**We fully agree with the reviewer on this point. We often neglected calcification as it is done by imperforate species, as they are not often used in paleoceanography. We have carefully assessed our abstract, introduction and discussion to emphasize that our results apply in principal to perforate foraminifera. For example, we added 'perforate' to lines 20 and 26 (abstract) and 61 (Introduction). We also refer to the potential difference between Globo- and Tubothalamea and added the two suggested papers on foraminiferal phylogeny in lines 256.**
**Section 4.2 describes the potential effect of the symbionts and their CA on our results. The reviewer is right, that species without (diatom) symbionts may react differently to incubation with AZ than *A. lessonii*. Therefore, we added a cautionary sentence at lines 225-226.**

*"Methods: There are many studies in the literature that discuss culture of Amphistegina spp. and Heterostegina depressa, as well as other benthic forams that host algal symbionts. The authors do not mention two important culture parameters, illumination (i.e., light intensities) and salinity. The latter may not be as critical to experimental results, since alkalinity is reported. However, light is a widely established, extremely important environmental parameter (e.g., Muller, 1978; Hallock, 1981; Hallock et al., 1986; Talge and Hallock, 2003; Williams and Hallock 2004)."*

**We thank the reviewer for pointing out this omission. Now, we added the light intensity at line 75: "Illumination was approximatively 180 µmol photons $m^{-2}$ $s^{-1}$, during the 12h of light"**
**The salinity was added to line 74: "They were fed with freeze-dried *Dunaliella salina* and incubated in North Atlantic sea water (salinity: 36)."**

*"In addition, the authors do not report the size range of individuals used in the experiments. If they had been aware of the earlier experimental studies, they would know that growth rates in Amphistegina are size dependent, which pertains to the comparability of their results to other studies (more on this in comments on the Discussion). Finally, in line 72, the authors mention that specimens were incubated in calcein prior to starting the experiment, with no mention of why and no further mention of calcien in the manuscript. Recommendations: Please report the light environment of the cultures, the salinity of the culture media, and the approximate starting size (or ending, since the experiments were very short) of the experimental specimens. Also, either elucidate on the use of calcein or delete mention of it."*

**The calcein was used to analyze whether the number of chambers formed matches the alkalinity decrease during the experiment. Results show that there is a consistent addition of new chambers with changes in total alkalinity: we now added a table with those results as supplementary information (Table S1).**
**We chose using individuals of the maximum possible size range in order to account for any size-specific responses. We agree with the reviewer and have added the range of starting diameters to the text (line 80).**

*"Results: The results are relatively straightforwardly presented. The only suggestion is that, in Table 1 and Table 2, reporting the decimal values for initial TA and initial DIC are not meaningful, given the standard deviations of the changes reflect whole numbers that represent ~10–40% of the changes in TA and DIC. "*

**We followed recommendation of the reviewer 1 and rounded all values in table 1 and Table 2.**

*"Discussion: See the comments and recommendations under "Introduction. That is, the Discussion should be focused on Amphistegina as a model for hyaline forams with algal endosymbionts, not all benthic forams."*

**We agree and have added a number of cautionary sentences to the discussion.**

*"Moreover, the authors state in lines 159–160, "The only previous study using Amphistegina spp.";
that statement is inaccurate. Ter Kuile and Erez (1984, 1987); and Hallock et al. (1986), all reported
rates of calcification in units equivalent to those reported. And indeed, the calcification rates reported
in the submitted manuscript are lower than most of the previously reported rates for Amphistegina,
which is why this reviewer questioned the light environment of the experiments. If the light levels
inside the culture flasks were limiting photosynthesis and growth of the experimental specimens, the
calcification rates would of course be relatively low. See, for example, Table 1 in Hallock et al.
(1986), who reported growth rates in μg/day dry weights at five different light intensities for both A.
lessonii and A. gibbosa. The growth rates at the lowest light levels are similar to those reported in the
submitted study. Moreover, the authors should note the starting diameters of the specimens used in
the Hallock et al. (1986) paper. The experiment reported in Table 1 in that paper used recently
produced juveniles, while the experiments reported in Table 3 included one trial with intermediate-
sized specimens (500–600 um diameter), while the other trials also used small juveniles. If the
experiments reported in the submitted paper used specimens in the 1–1.5 mm size range, the biology
of the forams indicates that only a few specimens would have added new chambers."*

**We agree with the reviewer and have re-phrased the end of the discussion's first paragraph
accordingly. We summarize the previously published growth rates to compare them to ours and
we added the suggested references.**

*"Lines 177–179: The authors suggest that calcification in Amphistegina might differ from that
reported in G. sacculifer, which is interesting, because, elsewhere, they are equating calcification in
Amphistegina with calcification in miliolids, which are far more distantly related, as noted above."*

**We see the reviewer's concern, which we now have hoped to have repaired by focussing the
discussion of our results to perforate foraminifera.**

*"In the paragraph in lines 187–196, the authors appear to assume that photosynthate produced by the
algal symbionts is primarily used for organic matrix. They do not consider the production of simple
sugars that can be used in ATP production that drives the ion pumps. In the case of Amphistegina,
TEM studies have shown the abundance of lipid storage bodies in the vicinity of the symbionts.
Indeed, the authors' conclusion that more research is needed on the types of organic molecules
produced is certainly true, but they overlooked pertinent information in papers by Lee, Stuhr, Talge,
Toler, and probably others. They also overlooked pulse-chase studies by Muller (1978) and ter Kuile
and Erez (1987)."*

**The suggestion that the symbionts are producing sugars for the foraminifer was added to line
197-199 including the suggested papers in this and following sentences. We extended the
discussion by citing reports on the positive effects of photosynthesis on calcification, which is in
line with our results.**

*"In lines 193–194, the authors mention "symbiotic dinoflagellates and zooxanthellae". Zooxanthellae
are symbiotic dinoflagellates."*

**We removed "and zooxanthellae" from this line.**

*"Conclusions: Of course, it is photosynthesis that enhances growth and calcification in
Amphistegina; and light is required for photosynthesis. The phylogenetic and physiological capability
to calcify is inherent in the hyaline forams (that is why Amphistegina can exhibit some calcification in
the dark). But because calcification is an energy-driven process, the substantial energy and organic
matter provided by photosynthesis by algal symbionts substantially enhances growth, including
calcification."*

**We fully agree that photosynthesis contributes to calcification by providing energy. However it was suggested that calcification and photosynthesis might be competing processes (Ter Kuile et al., (1989b, 1989a) which is contrary to what we observed. Therefore, we mentioned this in the conclusions.**

*"References: The references are typically "end-note" formatting-problematic and need extensive editing if Biogeoscience requires consistency in referencing."*

**We carefully edited the reference section.**

---

## Author Comment (AC2) · 25 Dec 2019

**Author responses to review of Anonymous Referee #2 to "Carbonic anhydrase is involved in benthic foraminiferal calcification"**
by Siham De Goeyse, Alice E. Webb, Gert-Jan Reichart, and Lennart J. de Nooijer

We are very grateful to the anonymous reviewer for the helpful comments on our manuscript. We reply below to the specific comments. Reviewer comments are given in italic font and our response in bold font.

**GENERAL COMMENTS**

*"The authors present a set of experiments, performed on a high number of foraminifera specimen of a same common symbiotic species. In this experimental work, the effect of an extracellular inhibitor of the carbonic anhydrase (CA) enzyme is compared to the effect of a photosynthesis inhibitor, as well as the sole light deprivation. Biomineralisation change is evaluated through measurements of DIC concentration and alkalinity change, and using solely that approach to evaluate "biomineralisation yield" is also a main aspect of the article. These results evidence, in my opinion, the role of carbonic anhydrase, but I do believe that additional simple information should be given in order to confirm that no other phenomenon can explain, or interfere with, those results. If those information can be given (see below) and the role of CA is confirmed, then the scientific significance of these results is excellent. The scientific quality is good, the method and experimental aspects are good despite the few information lacking, as great effort were provided to replicate the experiments and perform them on a significant amount of specimens. The discussion however and the manuscript text in general is not as good as I believe necessary for publication in an international journal. There are not enough references backing information, several aspects of the results are not discussed, a part of the discussion is just a description of results, there are words missing in some sentences, one name on a figure and a table do not match, one figure permitting the comparison of all results is missing, and there are several typography mistakes. I am not able to properly judge the english, but I found the manuscript perfectly understandable. If the text of the manuscript can be improved by the authors, I would recommend publication of the article as the results constitute a major advance in the understanding of biomineralisation by foraminifera (and in my opinion, it gives insight on biomineralisation mechanism in general considering how widespread is CA). For that reason, I hope the authors will improve the text, scientific content and discussion of the article in order to provide these interesting results the context they deserve to become a well referred to article."*

**We thank the reviewer for the kind words and helpful comments. We have critically assessed the text of our manuscript and listed our answers to all comments below. We hereby hope that our manuscript now is fit for publication in *Biogeosciences*.**

**SPECIFIC COMMENTS:**

*"I understand that solely using chemical solution parameters to describe the evolution of biomineralisation is one of the suggestion of the article, I however believe it is not enough as*

*some other parameters can affect DIC concentration and alkalinity: ex: microbial proliferation or open system phenomenon (improved gas /liquid phase exchanges in one experiment because of slightly different pH, or temperature differences due to the use of aluminum foil etc. . .). In my opinion the interest of a laboratory experiment on living organism cultured in vials is to be able to observe directly these organisms, which is not possible in other type of experiments. Are, in the end of the experiment, the vials clean enough with no particular microbial proliferation in one treatment? What proportion of foraminifera survived the experiment in all setups?"*

**We carefully monitored for signs of biofilm proliferation, of which there was no sign. In addition, there was no difference in appearance of the water, nor the vials between treatments, which is likely due to the relatively short incubation period and use of nutrient-poor seawater. We added the notion of the potential effect of biofilm proliferation to the end of the methods (lines 133-136).**

*"How are the new chambers? The author used calcein, they should thus be able to image the new chambers formed in each media. I believe any experiment of that type should present some kind of imaging, or at least a description of the visual aspects of the experiment, validating that new chambers formed, and evaluating that no microbial proliferation could have explained death of several microorganisms, that could explain less biomineralisation. For example, in the acetazolamide experiment, if a microbial proliferation occurred and a third of the foraminifera died, while the other survived and biomineralised regularly, wouldn't it give the impression, just by measuring DIC and alkalinity variation that only biomineralisation was affected by acetazolamide? These are simple information that would strengthen the results and the methods, that should be provided in the manuscript before publication. "*

**We agree that we cannot distinguish between 'normal chamber addition rates' by less specimens, and less chambers added by all specimens during the incubation after addition of AZ. We added this to the manuscript, although we would like to stress that both these options (less chambers by all specimens versus less specimens that added chambers) lead to largely the same conclusion. We now mention the number of chambers added to the supplementary information: these results also suggest that the same number of individuals added less chambers. By far most of the specimens added only one chamber after addition of AZ instead of two/ three in the control vials (now added as a supplementary table, S1). If half of the specimens would not have added chambers at all and the other half would have 'normal' chamber addition rates, more specimens would have been found with two/ three chambers added after addition of AZ. This implication has been added to the discussion (lines 190-192).**

*"In my opinion, authors should find a way to represent the results of all the different experiments together in one figure to ease comparison. As an example they could use the "corresponding g/L precipitated calcite" calculated for each experiment. "*

**We propose keep figures 3 and 4 separate, since there are (minor) experimental differences that may complicate a direct comparison. Length of the experiment and**

**initial size distribution of the foraminifera may have been slightly different and therefore may artificially increase differences between treatments.**

*"When discussing the effect of phostosynthesis on calcification (line 192) the author do not mention the effect of lowering ATP production and rather suggest that photosynthesis promotes the production of molecules that are used in organic templates of calcification. The role of ATP in chamber formation is, in my opinion, impossible to ignore, the author must discuss it in the manuscript. On the other hand, organics produced by the symbionts may help biomineralisation (this indeed need further investigation), but it should be mentioned that there are (many?) benthic foraminifera with a hyaline test that do not bear symbionts. This should be discussed by the authors as well."*

**We agree with the reviewer and, as replied to reviewer 1 too, we added this possibility to 4.2 (lines 216-220) in the revised version of our manuscript.**

*"Technical corrections Missing words or information: L29: "saturation state" the author should specify that it is towards calcium carbonate"*

**We changed this into "the saturation state of sea water with respect to calcite" and changed the sentence in L39 similarly: "Addition of $CO_2$ to sea water not only reduces saturation state with respect to calcite".**

*"L35-36: The authors could specify the foraminifera species (benthic ? planktonic ? Amphistegina ?)"*

**We added "benthic" here.**

*"L44: "Since this uptake. . .." The sentence sounds odd, a word is probably missing, it should be rephrased. "*

**We removed 'it' after 'and' from this sentence.**

*"L48: There is a dot after the bracket "*

**We removed it.**

*"L48: The sentence states "It was recently suggested that CO2 uptake by foraminifera is achieved through proton pumping" is that correct or did the authors used a shortcut to say that proton pumping (and thus ATP consumption) is used to modify pH and thus favor CO2 uptake/or that a proton pump is used to actively cotransport CO2? This imprecision should be corrected. "*

**It was indeed a shortcut to indicate that foraminifera use proton pumping to locally shift the pH around the site of calcification and thereby increase the $CO_2$ gradient and hence promotes inward $CO_2$ diffusion. We now extended this statement by explaining**

**the mechanism proposed by Toyofuku et al. (2017). This is added to lines 51-55 of the revised version of our manuscript.**

*"L50: In my understanding Bentov paper rather says that CO2 gets concentrated in low pH vesicles, and that from there it diffuses to the high pH vesicles where it converts into charged DIC species and is thus trapped inside the vesicle. The authors should clarify that point if I am not mistaking. "*

**We agree with the reviewer, but essentially, Bentov et al. (2009) and Toyofuku et al. (2017) describe the same carbon concentrating mechanism. They only differ in the location where this happens (i.e. either within vesicles or between SOC/ outside medium), but both propose differences in pH and thereby $p\text{CO}_2$ as a way to promote diffusion of $\text{CO}_2$ towards the location where calcification proceeds. We have extended the description of this process in the revised version of our manuscript (lines 52-55) to clarify this.**

*"Line 112: I am not familiar with the "T" symbol signification next to each alkalinity species, could it be clarified? "*

**The T was only here for "total". We have removed the 'T' from equations (1) and (2) as it was unclear.**

*"L115: Is that equation calculating "the alkalinity" or "the change in alkalinity", this should be clarified. "*

**We changed the text from "the observed change in alkalinity" to "the observed alkalinity" (now line 123).**

*Figure 4 caption: there is a dot after "represents"*

**We removed this dot.**

*L193: the sentence is missing a word*

**We changed this sentence into: "Utilization of photosynthate as an organic template for calcification may explain this observation" (now line 123 of the revised version of our manuscript).**

*"L198 : "it has been shown that symbiotic dinoflagellates and zooxanthellae can trigger the activity of carbonic anhydrase (CA) in their host organisms [. . .] thereby explaining how photosynthesis enhances calcification". The authors need to specify whether they mention the symbiont CA or if they refer to the host CA. Additionally the link between CA and photosynthesis must be explained. "*

**As we say in this sentence: '...of CA in their host organisms', although the reviewer may be right when hinting to enhanced activity of CA in the symbiont and thereby only indirectly helping calcification. Therefore, we have added the following sentence:**

**'Alternatively, increased activity of CA in the symbiont may also promote the flux of products to the host and thereby promote calcification indirectly.'**

**This section of the discussion has been altered according to other comments too, so that it now describes several possible interactions between CA, photosynthesis and calcification.**

*"Missing references: L27: "1/3rd of the carbon. . ." a reference should be provided"*

**We added a reference (Sabine and Tanhua, 2010).**

*"L40: Who suggested it? a reference is missing "*

**The reviewer is correct: there is no evidence for this, but the idea is sometimes brought up since other calcifyers are known to have such bicarbonate transporters. We therefore changed this sentence to avoid the suggestion that this uptake path has been shown to exist for foraminifera.**

*"L52: "many procaryotes and virtually all eukaryotes" a reference should be provided."*

**We added Lionetto et al., 2016 and Pastorekova 2004 to this sentence.**

*" L60: " a membrane impermeable inhibitor of this enzyme" a reference must be added. (A reference attesting that DCMU inhibits photosynthesis should be added as well if not provided in the manuscript). "*

**We added a reference to Moroney et al., 1985. Plant Physiol 79: 177-83. For the function of DCMU, we added a reference to Metz et al., 1986. FEBS Letters 205: 269.**

*"L207 : "Ca promotes [. . .] into the calcicoblastic space" this information should be supported by a reference"*

**We added a reference to Bertucci et al., 2013. Bioorg Med Chem 21: 1437-1450.**

*"L214: this reference and thus probably the whole sentence (except if another reference can be given) must be suppressed (as mentioned on Biogeosciences website "Works cited in a manuscript should be accepted for publication or published already"). "*

**We deleted this sentence.**

OTHER COMMENTS:

*"#1 Given that TA is measured with a 3µmol/kg precision and given the errors given in the tables, the decimals should be suppressed. "*

**We followed recommendation from reviewer 2 and reduced the number of decimals.**

*"#2 L78: What is the final concentration of dimethyl sulfoxide in the final flasks? The effect of dimethyl sulfoxide at that concentration on foraminifera should have been checked in a control experiment, if not, it should at least be discussed. "*

**The final concentration of DMSO was 0.05% (v/v) and the absence of impact had been tested on a preliminary experiment. We did not include these results since they would be merely a repetition of results shown by Moya et al. (2008), which we now added to line 85 of the revised version of our manuscript.**

*"#3 Figure 3 and 4, error bars should be represented or mentioned in the caption if smaller than symbols, or, even better, each 3 point replicates could be represented. "*

**We agree and have added values for ΔT$_A$ and ΔDIC for the individual measurements (in light gray). In case of the added 4, 8 and 16 µM AZ, we did not include them since the variability was very low: this is now indicated in the caption of figure 3.**

*"#4 Can the author explain why on figure 4 and 3 the control point is not at the same position (is there an explanation for these two different control results?). Additonnally, in table 1 and figure 3 there is two different names for one treatment, "No AZ" and "0 µM", please choose one wording."*

**The reviewer notes correctly that the averages for the two controls are not the same. This is due to the fact that results presented in figure 3 and 4 were obtained from experiments carried at two different time. Therefore, the initial size distribution of foraminifera was not the same. This explains why the 'control' vials gave different calcification rates. We also corrected the 'No AZ' from figure 3 to 'control, 0 µM'.**

*"#5 From line 159 to 170 it is a summary of the results that should not be, in my opinion, in the discussion. "*

**Here we respectfully disagree with the reviewer. It is indeed a summary of the results, but with no interpretation of them. Therefore, we propose to keep it where it was.**

*"#6 Line 175: The "extracellular" specificity of CA is mentioned here but not discussed in part 4.3 and then comes back later in the manuscript. This should be restructured to clarify the message of the authors. "*

**We agree and have specified where necessary 'extracellular' in 4.3.**

*"#7 Line 183: "the discrepancy between results may be caused by differences in the process involved in calcification between these species" Can the author mention one or more process they are referring to?"*

**We have included the following (now line 205-208):**

"For example, it has been suggested that calcification may involve seawater transport (Erez, 2003; Segev and Erez, 2006) as well as transmembrane transport (Nehrke et al., 2013; Toyofuku et al., 2017), of which the relative contribution may vary between groups of foraminifera."

---

## Author Response (AR1)

**List of relevant changes made in the manuscript:**

Dear editor, dear anonymous reviewers,

We have prepared a new version of our manuscript (BG-2019-356) according to the comments by the two reviewers. In addition to the previous answers, we have made a few more changes which we listed below. We hope that this adequately answers the concerns of the reviewers and that you will consider this new version for publication in Biogeosciences.

Sincerely,

Siham de Goeyse

Specific comments:

Title: has been changed into: "....by the benthic foraminifer *Amphistegina lessonii*."

Results: we have added the number of chambers added per treatment. This is added as a table and we refer to it at the end of 3.1 (line 176).

[revised manuscript text omitted]

---

## Author Response (AR2)

Dear editor, dear reviewers,

We thank you for the time spent on our manuscript. We have now prepared a new version of our manuscript (BG-2019-356) according to the comments made by the reviewers. Changes made to the text are detailed below.

Answer to Report #1

Submitted on 30 Jun 2020
Referee #3: Takashi Toyofuku

*Reviewers' comments are displayed in regular style whereas the author's reply is in italic bold.*

General comments:

I like this study very much.
This study is highly commendable because it an elaborate experimental design with reliable experimental techniques by well-established laboratory. The experimental results are robust. The results are novel and are of interest to many audiences

*We thank the reviewer for his kind words and constructive assessment. Below, we indicated how we changed our manuscript based on the comments*

Specific comments:

L78-79: Could the authors show the size distribution of each experimental condition in the supplemental materials?
*We added a figure to the supplementary material to show the size distribution per group. Since we do not know whether juvenile/ adult specimens respond differently to the treatments, we decided to incubate specimens large size range. This way we avoid a potential bias when extrapolating results to specimens from a specific size range. We now highlight this in the new version of our manuscript: line 81. : "After a week, viable specimens were collected and divided over eight experimental conditions, each of them consisting of three groups (Fig. 1). Each group consisted of 40-60 specimens with a similar size distribution (initial diameter: 140 to 1200 µm, shown in S1). "*

[Figure]

*S2: Size distribution of the individuals at the beginning of the experiment*

L94 Authors should indicate why they have quantified the amount of precipitation from DIC and alkalinity. It is clearly worth stating that there is no other way to estimate it precisely.

*We thank the reviewer for this suggestion and have added this precision at L96-98 : "**This method was chosen above other growth method measurement such as sample weighing (which is destructive) or chamber count as it allows a quantification of the amount of calcite formed during the experiment.**"*

L114 Authors should explain the meaning of the color of the arrows and the dotted lines in the captions. The same manner in later figures 3 and 4.

*We followed the reviewer's suggestion and added precisions to the description of the figure. See new graph and caption added below.*

[Figure]

*Figure 2: Calcification and net respiration of foraminifera deduced from changes in DIC and total alkalinity over time. The blue vectors show the impact of photosynthesis and respiration (impacting DIC), the red arrow show the impact of calcification and calcite dissolution (impacting both DIC and TA in a 1:2 ratio). Observed changes for each incubation should be decomposed into two vectors: a contribution of calcification (dashed red arrow) and the net effect of*

*respiration and photosynthesis (dashed blue arrow).*
*Approach is indicated here for a hypothetical incubation*

I would like to see the photos of individuals grown in the control, AZ, and DCMU conditions.
*SEM pictures of individuals grown under different conditions have been added to the supplementary information (figure S1).*

L184 As a previous reviewer pointed out, calcification is thought to require energy. It's hard to distinguish whether the problem is a shortage of energy or the insufficiency of photosynthesis itself.

Given the dark conditions every 12 hours, I'm not sure that the competition for carbon between photosynthesis and calcification is a problem. It should also be pointed out that sharing the CO2 by time may be occurring.
*We agree with the reviewer's comment and added this precision L202-204: "As the foraminifera were in the dark 12h hours a day it is feasible that DIC is shared over time, being used for calcification during the dark phase and for photosynthesis during the light phase. "*

For example, would calcification have been enhanced even if there were no dark conditions for 24-hour? Authors don't need to answer this question, but I think this sort of question would be helpful to sort out the really dominant factors
In the pre-experimental period when frozen algae were

given, how much numbers of chambers were added?

*We have not quantified number of chambers added in the time before starting the experiment. We assume that the number of chambers added during the pre-experimental period is similar to the number of chambers added during the experiment under "control" conditions (one or two chambers).*

L220 CA is an enzyme that is extremely universally found in the cytoplasm. I do not deny that CA is involved in calcification process, and I also believe so. However, the possibility that the activities of CAs of non-calcification site may also affect calcification. The possibility of widespread inhibition of the metabolic activity should be clearly described.

**We agree with the reviewer and have stressed this in the Discussion of our manuscript L236-237: ". It is also likely that cytoplasmic CAs -involved for instance in intracellular pH regulation- also affect calcification."**

Report #2

Referee #4: Anonymous referee

General comments:

A paper by de Goeyse et al. conducted a simple incubation experiment to test the role of carbonic anhydrase (CA) using the inhibitor acetazolamide on the calcification of symbiont-bearing foraminifer Amphistegina. Although the results clearly show the

involvement of CA on calcification of the foraminifer, it is still a vague impression to me where CA is present at the surface of cell membrane or the site of calcification. I look forward to authors' future cellular-scale studies to solve this question.

Some technical corrections are necessary prior to the acceptance of this paper.

*We thank the reviewer for the assessment of our manuscript: below, we answer point-by-point to the comments.*

Specific comments:

L11: Symbiont>symbiont
L17: seawater is taken up > how?
L38: Hikami et al., 2011)) > delete the last )
L51: SOC > site of calcification (SOC). Spell in full when first mentioned in the text
L54: the conversion from HCO3- > the conversion from HCO3- outside the test/cell membraned?

*We have changed the text of our corrected manuscript accordingly and thank the reviewer for pointing out these elements.*

L55: This process may be catalyzed by an enzymatic conversion by carbonic anhydrase (CA) > Does this process occur at SOC?
*This is indeed an important issue and unfortunately, we cannot solve this with the current dataset. As AZ is not*

*membrane-permeable we here hypothesize that the enzyme is located within and/or at the outer cell membrane.*

L79: similar size distribution (initial diameter: 140 to 1200 μm) > too broad initial diameter
*We did not want to suggest that the sizes of all incubated specimens were similar, but rather that the sizes (and size distribution) were similar between groups. We have changed the wording here and, also in reply to the first reviewer, added an additional figure showing the sizes of the incubated individuals.*

L103: (Liu et al., 2015) > Liu et al. (2015)
L119: the first one is constant the second present > the first one is constant and the second one is present

L144: many specimens in the control vials added 2 or 3 chambers > According to Table 3, most specimens added 1 or 2 chambers; only one specimen added 3 chambers.

L152: 42 and 16 > 16 and 42

L153: only 22 (resp. 19) $\mu mol \cdot L-1$ > only 19 and 22 $\mu mol \cdot L-1$, respectively
L163: approximately 65 $\mu mol \cdot L-1$ > Where does this figure comes from ?

L165: 60 $\mu g \cdot Ind.-1 \cdot day-1$; > Delete ;
L168: 0.3-6.6 > Add unit
L173: Fig. 2 > Fig. 3?
L196: (Ter Kuile et al., (1989b, 1989a) > Ter Kuil et al.

(1989a, 1989b), but no 1989a, b in References
L201-203, 210: Cited references are not listed in the
References.
L205: in known as > delete in
L221: In for example > For example
*Text and references have been corrected accordingly*

L256: not light, but photosynthesis itself promotes
calcification in perforate foraminifera. > better to say "…
in symbiont-bearing perforate foraminifera". I suggest
that you should pay more attentions to differences
between light and dark respirations to understand light-
enhanced calcification.
*We have changed the sentence L256 according to the
reviewer's suggestion*

Fig. 4 caption: Arrows show the calcification (red) and
net respiration (blue) effects. > Add this sentence in Fig.
2 and 3 as well.

*Figure and figure caption changed accordingly. New
caption now reads: " Calcification and net respiration
of foraminifera deduced from changes in DIC and total
alkalinity over time. The blue vectors show the impact
of photosynthesis and respiration (impacting DIC), the
red arrow show the impact of calcification and calcite
dissolution (impacting both DIC and TA in a 1:2 ratio).
Observed changes for each incubation should be
decomposed into two vectors: a contribution of*

*calcification (dashed red arrow) and the net effect of respiration and photosynthesis (dashed blue arrow). Approach is indicated here for a hypothetical incubation"*

Table 3: I am wondering if these specimens are dead or alive after incubation.
*Because of the limited during of the experiments (<5 days) (almost) all foraminifera were alive after the incubation. This has not been quantified, but no differences were observed between experiments. A preliminary experiment was performed prior to this study to make sure that the concentration of the inhibitor used did not affect foraminiferal survival. That experiment showed all foraminifera alive over a time span of a week.*